# Triage Strategies Based on C-Reactive Protein Levels and SARS-CoV-2 Tests among Individuals Referred with Suspected COVID-19: A Prospective Cohort Study

**DOI:** 10.3390/jcm11010201

**Published:** 2021-12-30

**Authors:** Erika Olivia Boyesen, Ida Maria Balsby, Marius Henriksen, Robin Christensen, Jens Henning Rasmussen, Finn Erland Nielsen, Hanne Nygaard, Lennart Jan Friis-Hansen, Susanne Dam Nielsen, Rebekka Faber Thudium, Celeste Porsberg, Lars Erik Kristensen, Henning Bliddal

**Affiliations:** 1The Parker Institute, Bispebjerg and Frederiksberg Hospital, 2000 Copenhagen, Denmark; erika.olivia.boyesen@regionh.dk (E.O.B.); ida.balsby@regionh.dk (I.M.B.); Marius.Henriksen@regionh.dk (M.H.); robin.christensen@regionh.dk (R.C.); lars.erik.kristensen@regionh.dk (L.E.K.); 2Research Unit of Rheumatology, Department of Clinical Research, University of Southern Denmark, Odense University Hospital, 5000 Odense, Denmark; 3Department of Emergency Medicine, Copenhagen University Hospital, Bispebjerg Frederiksberg Hospital, 2400 Copenhagen, Denmark; jens.henning.rasmussen@regionh.dk (J.H.R.); finn.erland.nielsen@regionh.dk (F.E.N.); hanne.nygaard@regionh.dk (H.N.); 4Department of Clinical Biochemistry, Bispebjerg Frederiksberg Hospital, 2400 Copenhagen, Denmark; lennart.jan.friis-hansen@regionh.dk; 5Department of Infectious Diseases 8632, Rigshospitalet, Copenhagen University Hospital, 2100 Copenhagen, Denmark; Susanne.Dam.Poulsen@regionh.dk (S.D.N.); rebekka.faber.thudium@regionh.dk (R.F.T.); 6Department of Pulmonary Medicine, Bispebjerg Frederiksberg Hospital, 2400 Copenhagen, Denmark; Celeste.Porsbjerg@regionh.dk

**Keywords:** COVID-19, C-reactive protein, CRP, triage strategies, SARS-CoV-2

## Abstract

C-reactive protein (CRP) has prognostic value in hospitalized patients with COVID-19; the importance of CRP in pre-hospitalized patients remains to be tested. **Methods**: Individuals with symptoms of COVID-19 had a SARS-CoV-2 PCR oropharyngeal swab test, and a measurement of CRP was performed at baseline, with an upper reference range of 10 mg/L. After 28 days, information about possible admissions, oxygen treatments, transfers to the ICU, or deaths was obtained from the patient files. Using logistic regression, the prognostic value of the CRP and SARS-CoV-2 test results was evaluated. **Results**: Among the 1006 patients included, the SARS-CoV-2 PCR test was positive in 59, and the CRP level was elevated (>10 mg/L) in 131. In total, 59 patients were hospitalized, only 3 of whom were SARS-CoV-2 positive, with elevated CRP (n = 2) and normal CRP (n = 1). The probability of being hospitalized with elevated CRP was 4.21 (95%CI 2.38–7.43, *p* < 0.0001), while the probability of being hospitalized with SARS-CoV-2 positivity alone was 0.85 (95%CI 0.26–2.81, *p* = 0.79). **Conclusions**: CRP is not a reliable predictor for the course of SARS-CoV-2 infection in pre-hospitalized patients. CRP, while not a SARS-CoV-2 positive test, had prognostic value in the total population of patients presenting with COVID-19-related symptoms.

## 1. Introduction

Coronavirus disease 2019 (COVID-19) is caused by severe acute respiratory syndrome coronavirus 2, SARS-CoV-2 [1]. The virus was initially discovered in Wuhan, China, in December 2019. In March 2020, the World Health Organization declared a COVID-19 pandemic, and, since then, the infection rate has increased exponentially worldwide [2]. As of December 2021, the coronavirus disease has resulted in over 276,400,000 confirmed cases and more than 5,374,700 deaths globally [3]. In Denmark, on 26 December 2021, 675,844 people had been infected, resulting in 3187 deaths in total [4]. COVID-19 can result in the development of acute respiratory distress syndrome (ARDS), which has high mortality [5]. ARDS is characterized by fibrin deposition, diffuse alveolar damage, with hyalin membrane formation and perivascular T-cell infiltration [6].

SARS-CoV-2 infection causes an acute phase response, partly driven by the pro-inflammatory cytokines interleukin-6 and TNF-a [7], which, again, induces a rise in the levels of acute phase proteins, including C-reactive protein (CRP). CRP was initially found to bind to the surface components (phosphorylcholine, PC) of pathogens, thereby activating the complement system and causing opsonization, leading to phagocytosis [8]. CRP also initiates the elimination of targeted cells by its interaction with both humoral and cellular effector systems of inflammation [9]. In clinical medicine, measurements of the plasma concentration of CRP are used as a multipurpose marker that, depending on the cut-off level used, can be applied, among other tests, to screen for inflammatory diseases and to differentiate bacterial from viral infections [10]. While other markers may be of value in specific patient groups, e.g., troponin [11], CRP has the advantage of being readily available in most clinics.

The lack of COVID-19 treatment resources threatens health systems worldwide. Costly biomarkers that can assist in the triage of COVID-19 patients are needed. Worsening of the virus infection, from the onset of symptoms to the development of ARDS, has been reported to be associated with a concomitant increase in CRP [12], and CRP has prognostic value in patients severely affected by COVID-19 [13]. Furthermore, the biomarker IL-6 has shown prognostic value regarding in-hospital deaths [14], and since the CRP synthesis in the liver is primarily driven by IL-6 [15], this supports the rationale behind investigating the role of isolated CRP in the early stages, for predicting the course of the disease. The present study included individuals with mild COVID-19-related symptoms, i.e., in the pre-hospital phase, when presenting at test centers in the Copenhagen area. We hypothesized that by using one simple biomarker (CRP), it would be possible to predict the likelihood of hospitalization and/or ARDS among suspected COVID-19 patients. For this, a CRP above the defined upper reference range of 10 mg/L at baseline was chosen as our pre-specified threshold of interest. Our aim was to explore whether there was potential for an effective triage strategy, based on the CRP result, to distinguish COVID-19 patients with an increased risk of hospitalization.

## 2. Materials and Methods

### 2.1. Study Design

This prospective cohort study was designed as a two-center study, carried out using prospective data from the corona check point (CCP) units at the Department of Acute Medicine, Bispebjerg Frederiksberg Hospital (BFH) and from the Department of Infectious Diseases, Rigshospitalet (RH) between May and July 2020. The study protocol was approved by the Danish Data Protection Agency (P-2020-395), while the project objective, outcomes, and outline were submitted to ClinicalTrials.gov NCT04373798 before enrolling participants. The pre-specified, approved, and registered protocol is available in Appendix A and the statistical analysis plan in Appendix A.

### 2.2. Setting and Participants

Individuals (aged 18 and above) presenting with suspected COVID-19 during daytime (8 a.m. to 5 p.m.) at the CCP units, whether referred from general practice (mainly BFH) or non-referred (mainly RH), were asked to participate in the prospective data collection. All participants presented with COVID-19-related symptoms, such as fever, muscle soreness, headache and coughing. Exclusion criteria were examination at one of the CCP units or hospitalization for the same diagnosis, within the past 28 days. 

All participants signed an informed consent form, had a blood sample drawn and a RT-PCR oropharyngeal swab specimen performed at the CCP. Participants were registered in the electronic medical record system and followed record perusal for one month (28 days). Data concerning test results, demographic details, possible hospitalizations, oxygen treatments, transfers to the ICU or deaths were registered in a dedicated database using the Research Electronic Data Capture software 10.6.18. (REDCap, Vanderbilt, TN, USA). In this study, time is measured in days, and *T* = 0 is the time of SARS-CoV-2 PCR testing, as the time of the onset of symptoms was uncertain.

### 2.3. Outcomes

The primary outcome was hospitalization (within the 28-day observation period) combined with CRP levels (above 10 mg/L) at baseline, in patients with suspected COVID-19. The secondary outcomes were transfer to the ICU, oxygen requirement and/or death (within the 28-day observation period) in combination with CRP levels (above 10 mg/L) at baseline, in patients with suspected COVID-19.

### 2.4. Data Extraction

Data extracted from the medical records included the CRP level, both at the time of appearance at the CCPs and at eventual later measurements, SARS-CoV-2 test results, and hospitalization. The data extraction included demographic information (sex and age), length of admission, intensive care, assisted respiratory treatments and death. The test for CRP was performed routinely at Rigshospitalet and Bispebjerg Frederiksberg Hospital.

### 2.5. Quantitative Variables

Initially, patients with a CRP above the clinical threshold (10 mg/L) were considered as having a “high CRP” (unlike those below the threshold—“low CRP”). Secondarily, the association between CRP and hospitalization was explored. Various thresholds, defined by data-driven categories that relate to hospitalization within 28 days from presenting at the CCPs with a suspicion of COVID-19, were used. For these exploratory purposes, a receiver operating characteristic (ROC) curve was used to graphically show the connection/trade-off between clinical sensitivity and specificity for every possible cut-off for the baseline CRP levels.

### 2.6. SARS-CoV-2 Detection and CRP Measurements

In this study, samples for detecting SARS-CoV-2 were collected by oropharyngeal swabs using the UTM swab set (COPAN, Brescia, Italy) followed by laboratory RT-qPCR analyses using E-gene assay [16] using Roche FLOW system (Roche, Basel, Switzerland). CRP was measured in heparin plasma using the c702 analyzer (Roche Diagnostics, Mannheim, Germany) [17]. The CV% was 7.5% at 7.6 mg/L. The upper limit for the reference value (non-specific indication) for CRP was 10 mg/L.

### 2.7. Statistical Methods

The primary analyses were based on the intention-to-survey (ITS) population. This ITS principle asserts the effect of the initial exposure (that is, the result of the SARS-CoV-2 and CRP level at enrolment). Thus, our ITS population was exclusively participants who signed the informed consent and had both CRP assessed and the result of SARS-CoV-2 available at baseline. Accordingly, participants were eligible for a prognostic group (e.g., high CRP vs. low CRP, and SARS-CoV-2 positive vs. negative test). They were followed up, assessed and analyzed as members of that group, regardless of their clinical history, from that point on (e.g., independent of withdrawals and cross-over phenomena) [18].

Logistic regression analysis models were used to describe data and to explain the relationship between the dependent binary variable(s) and the two independent variables (SARS-CoV-2 and CRP tests, respectively) [19]. Odds ratios (ORs) and 95% confidence intervals (CIs) were estimated. To compare differences between groups in each 2 × 2 contingency table, Fisher’s exact test was used; a two-tailed *p* value < 0.05 was considered statistically significant. All statistical analyses were performed using SAS software (Statistical Analysis Software 9.4, SAS Institute Inc., Cary, NC, USA).

## 3. Results

### 3.1. Clinical Characteristics of Patients

SARS-CoV-2 and CRP measurements were obtained from 1006 individuals (623 patients at BFH and 383 at RH) from May to July 2020; this sample represents our ITS population. As illustrated in Figure 1, only one patient was hospitalized in the group of SARS-CoV-2 positive patients with low CRP (n = 47), while two patients were hospitalized in the group of SARS-CoV-2 positive patients with high CRP (n = 12). In total, 59 patients were hospitalized within a mean time period of 6.5 days. None of the patients in the total ITS population were transferred to the ICU or died in the 28-day observation period. There were three SARS-CoV-2 negative patients who needed oxygen treatment during hospitalization, while none of the SARS-CoV-2 positive patients did.

In the total ITS population, the mean age was 45 years and 32.7% were male (Table 1). The individuals with a positive SARS-CoV-2 test (n = 59) had a mean age of 39.9 years and 17 (28.8%) of them were male. Among the SARS-CoV-2 negative patients (n = 947), the mean age was 45.3 years and 32.9% were male. 

### 3.2. Baseline CRP Levels and Clinical Characteristics 

In total, 59 patients (5.9%) were hospitalized. Three of them (5.1%) were SARS-CoV-2 positive and 21 of them (35.6%) had a CRP > 10 mg/L. Only two patients (3.2%) showed a combination of being SARS-CoV-2 positive, having increased CRP, and being hospitalized. The mean age of the hospitalized patients was 55.1 years and 49.2% were males. Three patients (5.1%), all SARS-CoV-2 negative, required oxygen treatment during their stay. The mean baseline CRP among the subsequently hospitalized patients was 39.4 mg/L (not shown), which is noticeably higher than the mean CRP in the other groups (Table 1). 

The mean CRP level among the SARS-CoV-2 positive patients (n = 59) was 9.1 mg/L, and 8.1 mg/L among the SARS-CoV-2 negative patients (n = 947). In the group of SARS-CoV-2 positive patients, the distribution of patients was 13.6%, 66.1%, and 20.3%, respectively, in the categories CRP < 1 mg/L, CRP between 1 mg/L and 10 mg/L, and CRP > 10 mg/L. In the group of SARS-CoV-2 negative patients, the percentages were 33.8%, 53.6%, and 12.6%, respectively (Table 1). 

### 3.3. Outcomes in Individuals with Suspected COVID-19

SARS-CoV-2 positivity alone had no predictive value regarding hospitalization in the unadjusted model (OR 0.85, 95%CI 0.26 to 2.81, *p* = 0.79). In contrast, patients with CRP levels above the clinical threshold (10 mg/L) at baseline had four times higher odds (unadjusted) of being hospitalized than patients with CRP levels below the threshold (OR 4.21, 95%CI 2.38 to 7.43, *p* < 0.001), meaning that CRP had significant predictive value concerning the hospitalization of patients with suspected COVID-19 (Table 2). In total, 131 patients had high CRP at baseline and 16% of them (n = 21) were admitted during the 28-day observation period. Only 4.3% (n = 38) of the patients admitted had low CRP (not shown).

Among the patients with high CRP (>10 mg/L), male sex and higher age (in years) corresponded with a higher risk of being hospitalized (Table 2) in the unadjusted model. Differences between hospitals in the number of hospitalizations were noticeable; patients at RH were 69% less likely to be hospitalized (OR 0.31, 95%CI 0.16 to 0.63, *p* = 0.001), according to the unadjusted model, than patients at BFH. Oxygen requirement was not significantly associated with high CRP, sex, SARS-CoV-2 positivity, or hospital.

### 3.4. Hospitalizations and Risk Factors

SARS-CoV-2 positive patients with high CRP were associated with a higher risk of hospitalization (OR 9.20, 95%CI 0.76–111.63) in the unadjusted model, but the association did not show any significance (*p* = 0.08). In the adjusted model (multiple regression), the likelihood of being hospitalized was equal (OR 1.00) in the two groups (high and low CRP). Sex, age, and hospital were not significantly associated with hospitalizations among SARS-CoV-2 positive patients (Table 3).

### 3.5. Receiver Operating Characteristic Curve

As illustrated in Figure 2, we used the receiver operating characteristic (ROC) curve to show sensitivity as a function of 100%—specificity, determining a CRP threshold of hospitalizations among patients with suspected COVID-19, derived from the observed data. Using Youden’s index, the best predictive CRP cut-off level was 6 mg/L, with a sensitivity of 49.2% and a specificity of 82.2%. However, the area under the curve (AUC) was 0.66 (CI 0.58–0.74), indicating only a poor degree of class separability, as well as predictability of hospitalization.

## 4. Discussion

Although the clinical triage value of CRP is important, this study did not demonstrate a triage value of CRP in subjects with suspected COVID-19. However, in patients initially presenting (at baseline) with symptoms of SARS-CoV-2 infection, we found that high CRP was more likely to predict hospitalization in the total ITS population than low CRP (OR 4.21, 95%CI 2.38–7.43, *p* < 0.001), whereas SARS-CoV-2 positivity did not show any predictive value regarding hospitalizations. This knowledge may be useful in future diagnostics of patients presenting with mild signs of illness. In this material, a lack of significance for a similar association between CRP > 10 mg/L at the onset of symptoms and subsequent hospitalization among patients infected with SARS-CoV-2 was revealed. This may be due to a type 2 error; thus, the odds of being admitted to a hospital within 28 days from the baseline were 9.2 times higher in the group with high CRP compared to the group with low CRP (OR 0.92, 95%CI 0.76–111.63, *p* = 0.08). None of the participants in our study were affected to an extent that led to ARDS, and no conclusions can be made on the risk of being transferred to the ICU, requiring oxygen treatment, or dying (secondary outcomes) due to SARS-CoV-2 infection. 

### 4.1. Handling of SARS-CoV-2 

The current recommendations for the management of COVID-19 include large-scale antigen/PCR testing for SARS-CoV-2. While such tests reveal whether an individual is infected with SARS-CoV-2 or not, the demonstration of the virus has, to our knowledge, no prognostic value for the ensuing course of the COVID-19 disease. In Denmark, due to a high SARS-CoV-2 PCR and antigen testing capacity, the positive rate of the test has varied from 0.2 to 7% at our hospitals. This is also reflected in this study population, which had a very high percentage of negative tests, as the samples were collected in the last part of the first wave. During periods with higher positive test rates, there is a need for other measures to be taken, in order to identify the subjects who are at an increased risk of developing COVID-19 complications, which has proved to be effective in other respiratory diseases [20]. 

A cut-off value of 6 mg/L for CRP (regardless of SARS-CoV-2 positivity) was determined using the ROC curve, which is slightly lower than the official reference range of CRP, below 10 mg/L. This corresponded well with that observed with SARS-CoV-2 and MERS, as well as the four coronaviruses that cause milder disease. In hospitalized patients with COVID-19, similar ROC curves have suggested a cut-off for death from the disease of 41.4 CRP [13].

### 4.2. CRP in the Different Stages of COVID-19 Infections

In a recent systemic review [21], it was found that CRP levels are significantly higher in severe COVID-19 patients than in non-severe patients. This meta-analysis intended to deliver information to help physicians to distinguish between severe and non-severe COVID-19 patients. Indeed, in hospitalized patients, CRP has prognostic value, even in moderately affected COVID-19 patients [22,23,24]. However, no prognostic value of CRP in pre-hospitalized patients was described. In this study, CRP was not a significant predictor of the course of the COVID-19 disease, which indicates that the cytokine response and subsequent rise in CRP occur in later stages of the disease. It may be speculated that the immune reaction leading to CRP production only evolves from a certain level of disease, which corresponds to the worse prognosis in hospitalized patients with very high CRP levels. In severe COVID-19 cases, the serum concentrations of CRP are increased in non-survivors compared to non-severe COVID-19 patients and survivors [21]. Accordingly, CRP levels may be of value for monitoring the COVID-19 disease, once developed. 

### 4.3. Reflections on Investigation 

The 69% reduced risk of being admitted to Rigshospitalet, in comparison to Bispebjerg Frederiksberg Hospital, indicates a difference in the patient groups at the two hospitals, especially considering that the mean CRP was 6.6 mg/L higher in the SARS-CoV-2 positive patients and 4.8 mg/L higher in the SARS-CoV-2 negative patients at Bispebjerg Frederiksberg Hospital, compared to Rigshospitalet. This may reflect a more unwell group of patients at BFH, having mostly been referred to the CCP’s by their general practitioner (GP), as compared to the mostly non-referred patients arriving at RH. 

This investigation had several limitations. Only a minority of the participants were infected with SARS-CoV-2, which limited the relevance of the secondary outcomes, e.g., the risk of being transferred to the ICU, requiring oxygen, or dying due to COVID-19. The strength of the study was the inclusion of patients in the pre-hospital phase, as this group is only vaguely described in the literature. Moreover, the addition of a simple and cheap test, such as CRP, would provide the GP with a rapid tool for primary triage; the aim of this being to choose the right candidates for more intensive follow-up after the test, or even referral to the hospital. Thus, CRP may be one of the prognostic tools that predicts the morbidity of the COVID-19 course, but only in severe cases. 

## 5. Conclusions

In conclusion, our findings did not support the theory of CRP as a reliable predictor for the course of the SARS-CoV-2 infection in pre-hospitalized patients. The mean CRP was, in fact, slightly higher among SARS-CoV-2 positive patients than SARS-CoV-2 negative patients, although there was no significant association between CRP among SARS-CoV-2 positive patients and hospitalizations. CRP had prognostic value in the total population presenting with COVID-19-related symptoms (general signs of illness), while a SARS-CoV-2 positive test per se did not influence the outcome. 

## Figures and Tables

**Figure 1 jcm-11-00201-f001:**
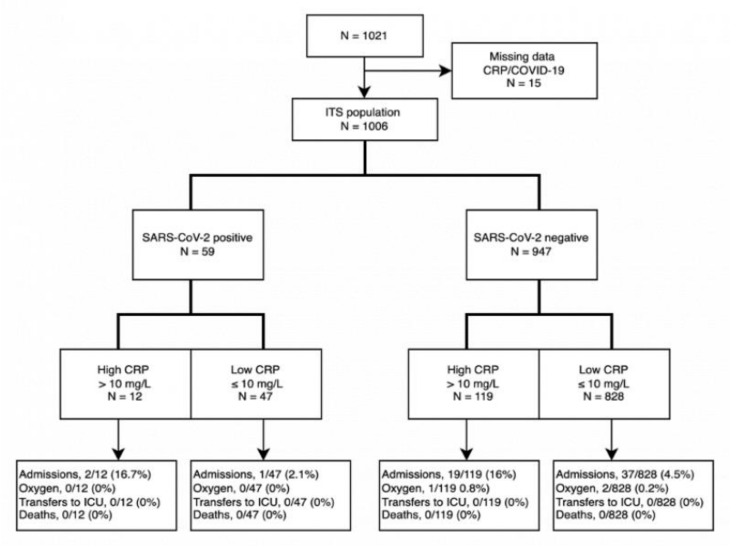
Flow of participants through each stage of the prospective cohort study.

**Figure 2 jcm-11-00201-f002:**
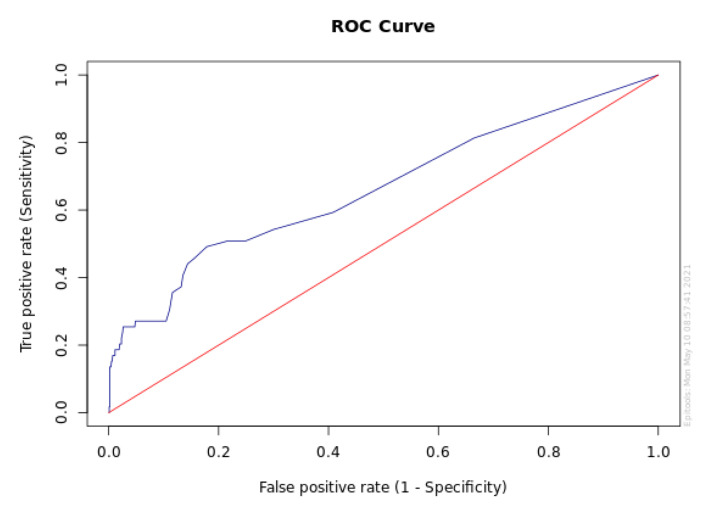
ROC curve with sensitivity as a function of 100%—specificity. CRP level vs. hospitalizations.

**Table 1 jcm-11-00201-t001:** Participant characteristics at baseline stratified according to subsequent SARS-CoV-2 test results.

SARS-CoV-2 Result	BFH		RH		Total		Overall Total
	Positive n = 37	Negative n = 586	Positive n = 22	Negative n = 361	Positive n = 59	Negative n = 947	All Participant n = 1006
Males, no. (%)	10 (27)	220 (37.5)	7 (31.2)	92 (25.5)	17 (28.8)	312 (32.9)	329 (32.7)
Age, years (mean)	39.7	46.5	40.3	43.3	39.9	45.3	45.0
CRP level, mg/L (mean)	11.6	9.9	5.0	5.1	9.1	8.1	8.1
CRP < 1 mg/L, no. (%)	2 (5.4)	134 (22.9)	6 (27.3)	186 (51.2)	8 (13.6)	320 (33.8)	328 (32.6)
CRP between 1 mg/L and 10 mg/L, no. (%)	26 (70.2)	367 (62.6)	13 (59.0)	141 (39.1)	39 (66.1)	508 (53.6)	547 (54.4)
High CRP > 10 mg/L, no. (%)	9 (24.3)	85 (14.5)	3 (13.6)	34 (9.4)	12 (20.3)	119 (12.6)	131 (13.0)
SARS-CoV-2 positive, no. (%)	37 (100)	0 (0)	22 (100)	0 (0)	59 (100)	0 (0)	59 (5.9)
High CRP + SARS-CoV-2 positive, no. (%)	9 (24.3)		0 (0)		9 (15.3)		9 (0.9)

**Table 2 jcm-11-00201-t002:** Odds ratios (95% confidence intervals) and *p* values for the primary and secondary outcome in individuals with suspected COVID-19 (n = 1006).

Outcomes ^1^	Odds Ratio (95%CI; *p* Value)		
	Simple Logistic Regression	Logistic Regression (With 2 Independent Variables)	Multiple Regression Analysis (Adjusted for the Other Variables)
Y: Hospitalization			
High CRP (>10 mg/L)	4.21 (2.38–7.43; *p* < 0.0001)	4.26 (2.41–7.53; *p* < 0.0001)	3.48 (1.91–6.33; *p* < 0.0001)
SARS-CoV-2 positive	0.85 (0.26–2.81; *p* = 0.79)	0.72 (0.22–2.43; *p* = 0.60)	0.81 (0.23–2.92; *p* = 0.75)
X1: Male sex	2.09 (1.23–3.54; *p* = 0.006)		1.59 (0.91–2.76; *p* = 0.10)
X2: Age in years	1.04 (1.03–1.06; *p* < 0.0001)		1.04 (1.02–1.05; *p* < 0.0001)
X3: Hospital ^2^	0.31 (0.16–0.63; *p* = 0.001)		0.42 (0.20–0.85; *p* = 0.02)
Y: Oxygen treatment			
High CRP (>10 mg/L)	3.36 (0.30–37.29; *p* = 0.32)	3.50 (0.32–38.90; *p* = 0.31)	2.85 (0.25–33.01; *p* = 0.40)
SARS-CoV-2 positive	n.a. (<0.98)	n.a. (*p* = 0.98)	n.a. (<0.98)
X1: Males	4.14 (0.37–45.76; *p* = 0.25)		3.65 (0.31–42.33; *p* = 0.30)
X2: Age in years	1.02 (0.96–1.10; *p* = 0.49)		1.02 (0.95–1.09; *p* = 0.58)
X3: Hospital^2^	0.81 (0.07–8.99; *p* = 0.87)		1.18 (0.10–14.10; *p* = 0.89)

^1^ No data were registered on transfers to the ICU or deaths. ^2^ Rigshospitalet vs. Bispebjerg Frederiksberg Hospital.

**Table 3 jcm-11-00201-t003:** Odds ratios (95% confidence intervals) and *p* values for hospitalizations stratified by the results of the SARS-CoV-2 test.

Outcome Stratified by SARS-CoV-2 Test	Odds Ratio (95%CI; *p* Value)	
	Simple Regression Analysis	Multiple Regression Analysis
Hospitalization among the SARS-CoV-2 positive individuals (N = 59)		
High CRP (>10 mg/L)	9.20 (0.76–111.63; *p =* 0.08)	1.00 (0.01–70.50; *p =* 0.99)
X1: Males	5.47 (0.46–64.77; *p =* 0.18)	3.91 (0.20–74.77; *p =* 0.37)
X2: Age in years	1.11 (1.02–1.21; *p =* 0.02)	1.09 (0.97–1.23; *p* = 0.16)
X3: Hospital ^1^	n.a. (0.96)	n.a. (0.96)
Hospitalization among the SARS-CoV-2 negative individuals (N = 947)		
High CRP (>10 mg/L)	4.06 (2.25–7.34; *p* < 0.0001)	3.47 (1.88–6.41; *p* < 0.0001)
X1: Males	1.98 (1.15–3.41; *p* = 0.01)	1.53 (0.87–2.69; *p* = 0.14)
X2: Age in years	1.04 (1.02–1.06; *p* < 0.0001)	1.03 (1.02–1.05; *p =* 0.0001)
X3: Hospital ^1^	0.34 (0.17–0.67; *p =* 0.002)	0.44 (0.21–0.90; *p* = 0.02)

^1^ Rigshospitalet vs. Bispebjerg Hospital.

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
