# Peer review of "Triage Strategies Based on C-Reactive Protein Levels and SARS-CoV-2 Tests among Individuals Referred with Suspected COVID-19: A Prospective Cohort Study"

_jcm, 2021, doi:10.3390/jcm11010201_

Round 1

Reviewer 1 Report

This study investigated whether the CRP level could be a reliable predictor of the course of SARS-CoV-2 infection in pre-hospital patients.

In my opinion the paper is worth studying and the manuscript contains enough original and interesting material. It is clearly and concisely written. The experimental procedures are described comprehensively. The results are well analyzed.

Minor corrections:

  1. The quality of figure and tables must be improved.
  2. Text formatting should be carefully checked.
  3. The language should be modified carefully.

Author Response

jcm-1478877

We should like to thank the reviewers for their perusal of the manuscript. Below please find answers to the queries and suggestions by the reviewers.

Also, included please find the revised manuscript with and without tracked changes.

Reviewer no 1:

  1. This study investigated whether the CRP level could be a reliable predictor of the course of SARS-CoV-2 infection in prehospital patients. In my opinion the paper is worth studying and the manuscript contains enough original and interesting material. It is clearly and concisely written. The experimental procedures are described comprehensively. The results are well analyzed.

Answer:

We thank the reviewer for this comment

Minor corrections:

  1. The quality of figure and tables must be improved.

Answer: The figure 1 and tables have been redrawn/reformatted.

  1. Text formatting should be carefully checked.

Answer: The text has been reformatted

  1. The language should be modified carefully.

English language has been revised according to corrections from a native English speaking person

Reviewer no 2:

  1. The manuscript ID jcm-1478877 entitled"Triage strategies basedon C-reactive protein levels and SARS-CoV-2 2 tests amongindividuals referred with suspected COVID-19: A prospective cohort study" is an interesting study in the aspect of COVID-19biomarkers.

Answer:

We thank the reviewer for this comment

  1. C-reactive protein (CRP) has a prognostic value in hospitalized patients with 30 COVID-19, the importance of CRP in the pre-hospitalized patients remains to be tested. Individuals with symptoms of COVID-19 had a SARS-CoV-2-PCR oropharyngealswab test and measurement of CRP was performed at baselinewith an upper reference range of 10mg/L. After 28 days, information was extracted from files about possible admissions,oxygen treatments, transfers to the ICU, or deaths. Using logistic regression, the prognostic value of CRP and SARS-CoV-2 test results was evaluated. In the results, among the 1,006 patients included, the SARS-CoV-2-PCR test was positive in 59, and theCRP level was elevated (>10mg/L) in 131. In total, 59 patients were hospitalized, only 3 of whom wereSARS-CoV-2 positive with elevated CRP (n=2) and normal CRP(n=1). Odds for getting hospitalized 39 with elevated CRP were4.21 (95%CI 2.38 – 7.43, p < 0.0001), while SARS-CoV-2-positivity alone gave 40 0.85 (95%CI 0.26 – 2.81, p = 0.79). Finally, the authors concluded CRP is not a reliable predictor forthe course of the SARS-CoV-2 infection in pre-hospitalizedpatients. However, CRP had a prognostic value in the total population presenting with COVID-19 related symptoms. Further, the following questions may be addressed before submitting a revision,

Answer:

We agree with the synthesis of the reviewer

  1. The following statement from the current study or literature info? "SARS-CoV-2 infection causes an acute phase response partly driven by the pro-inflammatory cytokines interleukin-6 andTNF-a, which again induces a rise in levels of acute-phase proteins 59 including C-reactive protein (CRP)" -----please cite properly.

Answer:

We have added several references to the text and changed accordingly

  1. How did the authors were selected CRP for this study? Why did they choose this marker?

Answer:

CRP is the most used indicator for acute phase reactants. This is possibly due to its accessibility and low cost. In our Region, Point-of-Care equipment is readily available. This has been noted in the text.

  1. There are studies and reviews reported that CRP is used for diagnostic and prognostic marker----PMID: 32516845, PMCID:PMC7928982, https://doi.org/10.3389/fimmu.2021.720363, and https://doi.org/10.1016/j.medmal.2020.03.007------but the current results are inverse----the author might be explained in detail

Answer:

We thank the reviewer for pointing out these articles. They deal with patients admitted to hospital with COVID-19 infection, while our material consisted of subjects tested in the prehospital setting. It may be speculated that the immune reaction leading to CRP production only evolves from a certain level of disease, which corresponds to the worse prognosis in hospitalized patients with very high CRP levels. A note of this has been added to both introduction and discussion with reference to the literature suggested.

  1. What is the ratio of males and females included in this study?

Answer:

Male and female ratio is given in the text (line 175) and in table 1

  1. Citations are missing in the biochemical analysis and statistical analysis section

Answer:

References have been added

  1. Table 1 legends are missing

Answer:

Legends of figures and tables are given in the revised manuscript when embedded in the text while the paragraph with supplementary material has been omitted.

  1. The authors may revise the conclusion appropriately

Answer:

The conclusion has been rephrased.

Reviewer no 3:

  1. This article wants to demonstrate that C-reactive protein (CRP)has a prognostic value in hospitalized patients with COVID-19because the importance of CRP in pre-hospitalized patientsremains to be tested. The conclusions document an important value of CRP but as an acute inflammatory state not necessarilyrelated to Sars-Cov-2.The information extracted from this article is certainly of greatimportance, to identify prognostic markers of the disease.In this sense I recommend reading and citing if possible thefollowing articles on the prognosis of Covid-19 disease:PMID: 32986136The level of English vocabulary is fairlyaccurate despite some revisions in the use of scientific languagewhich I consider appropriate.

Answer:

We thank the reviewer for the comment. As stated in the answer to reviewer no 1, the English text has been corrected. As for the Arcari reference, this deals with hospitalized patients and has been quoted in the paragraph added on the suggestion on reviewer no. 2 (e).

Reviewer 2 Report

The manuscript ID jcm-1478877 entitled"Triage strategies based on C-reactive protein levels and SARS-CoV-2 2 tests among individuals referred with suspected COVID-19: A 3 prospective cohort study" is an interesting study in the aspect of COVID-19 biomarkers. 

C-reactive protein (CRP) has a prognostic value in hospitalized patients with 30 COVID-19, the importance of CRP in the pre-hospitalized patients remains to be tested. Individuals with symptoms of COVID-19 had a SARS-CoV-2-PCR oropharyngeal swab test and measurement of CRP was performed at baseline with an upper reference range of 10mg/L. After 28 days, information was extracted from files about possible admissions, oxygen treatments, transfers to the ICU, or deaths. Using logistic regression, the prognostic value of CRP and SARS-CoV-2 test results was evaluated. In the results, among the 1,006 patients included, the SARS-CoV-2-PCR test was positive in 59, and the CRP level was elevated (>10mg/L) in 131. In total, 59 patients were hospitalized, only 3 of whom were
SARS-CoV-2 positive with elevated CRP (n=2) and normal CRP (n=1). Odds for getting hospitalized 39 with elevated CRP were 4.21 (95%CI 2.38 – 7.43, p < 0.0001), while SARS-CoV-2-positivity alone gave 40 0.85 (95%CI 0.26 – 2.81, p = 0.79).  Finally, the authors concluded CRP is not a reliable predictor for the course of the SARS-CoV-2 infection in pre-hospitalized patients. However, CRP had a prognostic value in the total population presenting with COVID-19 related symptoms. Further, the following questions may be addressed before submitting a revision,

1) The following statement from the current study or literature info? "SARS-CoV-2 infection causes an acute phase response partly driven by the pro-inflammatory cytokines interleukin-6 and TNF-a, which again induces a rise in levels of acute-phase proteins 59 including C-reactive protein (CRP)" -----please cite properly. 

2) How did the authors were selected CRP for this study? Why did they choose this marker?

3) There are studies and reviews reported that CRP is used for diagnostic and prognostic marker----PMID: 32516845, PMCID: PMC7928982, https://doi.org/10.3389/fimmu.2021.720363, and https://doi.org/10.1016/j.medmal.2020.03.007------but the current results are inverse----the author might be explained in detail

4) What is the ratio of males and females included in this study? 

5) Citations are missing in the biochemical analysis and statistical analysis section

6) Table 1 legends are missing

7) The authors may revise the conclusion appropriately. 

Author Response

(The authors gave the same response as above.)

Reviewer 3 Report

This article wants to demonstrate that C-reactive protein (CRP) has a prognostic value in hospitalized patients with COVID-19 because the importance of CRP in pre-hospitalized patients remains to be tested. The conclusions document an important value of CRP but as an acute inflammatory state not necessarily related to Sars-Cov-2.
The information extracted from this article is certainly of great importance, to identify prognostic markers of the disease.
In this sense I recommend reading and citing if possible the following articles on the prognosis of Covid-19 disease:
PMID: 32986136The level of English vocabulary is fairly accurate despite some revisions in the use of scientific language which I consider appropriate. 

Author Response

(The authors gave the same response as above.)

Round 2

Reviewer 2 Report

Accept after minor revision (corrections to minor methodological errors and text editing)